# A Meta-Analysis and Systematic Review of the Effect of Chatbot Technology Use in Sustainable Education

Xinjie Deng  and Zhonggen Yu *

Faculty of Foreign Studies, Beijing Language and Culture University, Beijing 100083, China
* Correspondence: yuzhonggen@blcu.edu.cn

**Abstract:** The development of artificial intelligence in recent years has promoted the use of chatbot technology in sustainable education. Many studies examined the effect of chatbots on learning outcomes. However, scant studies summarized the effectiveness of chatbots in education. The aim of the study is to investigate the effect of chatbot-assisted learning on various components and how different moderator variables influenced the effectiveness. This study, through a meta-analysis, reviewed 32 empirical studies with 2201 participants published between 2010 and 2022. The results showed that chatbot technology exerted a medium-to-high effect on overall learning outcomes regardless of moderator variables, i.e., intervention duration, chatbot roles, and learning content. In terms of learning components, chatbots could significantly improve explicit reasoning, learning achievement, knowledge retention, and learning interest despite negative findings in critical thinking, learning engagement, and motivation. Future research could expand chatbot research by including different learning components.

**Keywords:** chatbot technology; meta-analysis; learning outcomes; chatbot-assisted learning; sustainable education

## 1. Introduction

As the popularity of information technologies grows, chatbots have caught the increasing attention of stakeholders in the educational context. A chatbot is a conversational program that can process input and accordingly provide information through verbal or written interactions [1]. Researchers and practitioners could even design chatbots by themselves based on Dialogflow (e.g., [2,3]). Dialogflow is a natural language understanding platform integrating conversational interfaces into various devices, applications, and bots [4]. During the post-pandemic period when online learning still plays an important role, chatbot integration alleviates teachers' workload to provide individual support for students with limited resources and personalizes students' pace of learning [5]. Unlike teachers, educational chatbots could answer students' questions anytime and anywhere. They have the ability to handle several questions at the same time.

However, the use of chatbot technology also brings challenges. They can be described as issues in ethics, evaluation, users' attitudes, programming, supervision, and maintenance [6]. Problems have also included technological limitations and training side effects [7]. Simultaneously, there comes another issue. The novelty effect may appear when students are introduced to new technology. The improvement of learning outcomes might result from students' newness to chatbot technology [8]. Given the benefits and concerns of educational chatbot use, many studies have measured the effectiveness of chatbots but have obtained inconsistent findings.

Although some review studies focused on chatbot-based education, few of them synthesized previous studies to identify the overall effect of chatbots. Recent review articles either provided basic information on chatbot research through visualization (e.g., [9]), or summarized chatbot roles, evaluation methods, application fields, affordances, and

challenges through content analysis (Table 1). In terms of the effectiveness of chatbots, there were two meta-analyses calculating the overall effect size of chatbots on language learning. Lee and Hwang [10] limited their research to English as a foreign language education, and Bibauw et al. [11] also used language learning as the research topic.

**Table 1.** The research focus of relevant review studies.

| N | Focus | Relevant Review Study |
|---|---|---|
| 1 | Research authors, journals, and countries | Hwang and Chang [9] |
| 2 | Advantages and challenges of chatbot use | Huang et al. [1], Okonkwo, and Ade-Ibijola [6], Perez et al. [7] |
| 3 | Chatbot roles | Kuhail et al. [9] |
| 4 | Evaluation methods | Perez et al. [4], Kuhail et al. [12] |
| 5 | Application fields | Okonkwo and Ade-Ibijola [6], Hwang and Chang [9], Kuhail et al. [12] |
| 6 | The overall effect of chatbots on language learning | Lee & Hwang [10], Bibauw et al. [11] |
| 7 | The overall effect of chatbots on education | This study |

However, some previous studies only investigated a particular context. Lee and Hwang [10] focused on the effect of chatbots in the Korean context. In recent years, the Chinese Ministry of Education has been advocating technology-enhanced education. Chatbots could improve Chinese students' thinking ability and facilitate interactive learning [13]. The meta-analysis of chatbot technology in education across the world remains sparse. This study thus aims to examine the effect of chatbot use on various components and whether the effectiveness would be influenced by different variables. This study would shed light on the effect of chatbot-assisted learning not only in China but also in other countries and regions. It could also provide a reference for sustainable education and developing certain abilities and affective domains.

## 2. Literature Review

Chatbots perform three roles in education, i.e., teaching assistants, learning partners, and personal tutors. Inspired by Li and Yu [14], the authors summarized the role of chatbots in Figure 1. Operating as a teaching assistant, the chatbot mechanism provided professional knowledge and formative feedback [15] and scaffolded students' online learning [16]. Chatbots, as learning partners, chatted and interacted with students through either texts or voices. The tutorial role required chatbots to offer questions and answers, guided students to start their learning [17], and give quizzes [18]. The three educational roles of chatbots are intertwined with each other, contributing to effective teaching and learning [14,19]. Given the function of chatbots, it is most likely that chatbot-based education would positively and significantly influence critical thinking, explicit reasoning, learning achievement, knowledge retention, engagement, motivation, and interest.

### 2.1. Critical Thinking

Critical thinking refers to the thinking process of forming self-regulatory and reflective judgments which could determine one's beliefs and behaviors [20]. As one of the 21st-century skills, critical thinking has become increasingly pivotal in education [21]. Students are encouraged to express their viewpoints based on critical analysis and reasoning [22]. Therefore, recent years have witnessed many studies on the cultivation of critical thinking, especially with the assistance of artificial intelligence and information communication technology tools [23]. The use of emerging technologies such as chatbot systems could guide and inspire students to think over and make judgments, thus gradually developing the habit of critical thinking. The artificial intelligence-integrated chatbot was proved effective to enhance students' thinking ability and expectations [13]. Accordingly, the authors proposed the following null hypothesis.

**H1.** *The use of chatbot technology could not significantly improve critical thinking at the 0.05 level.*

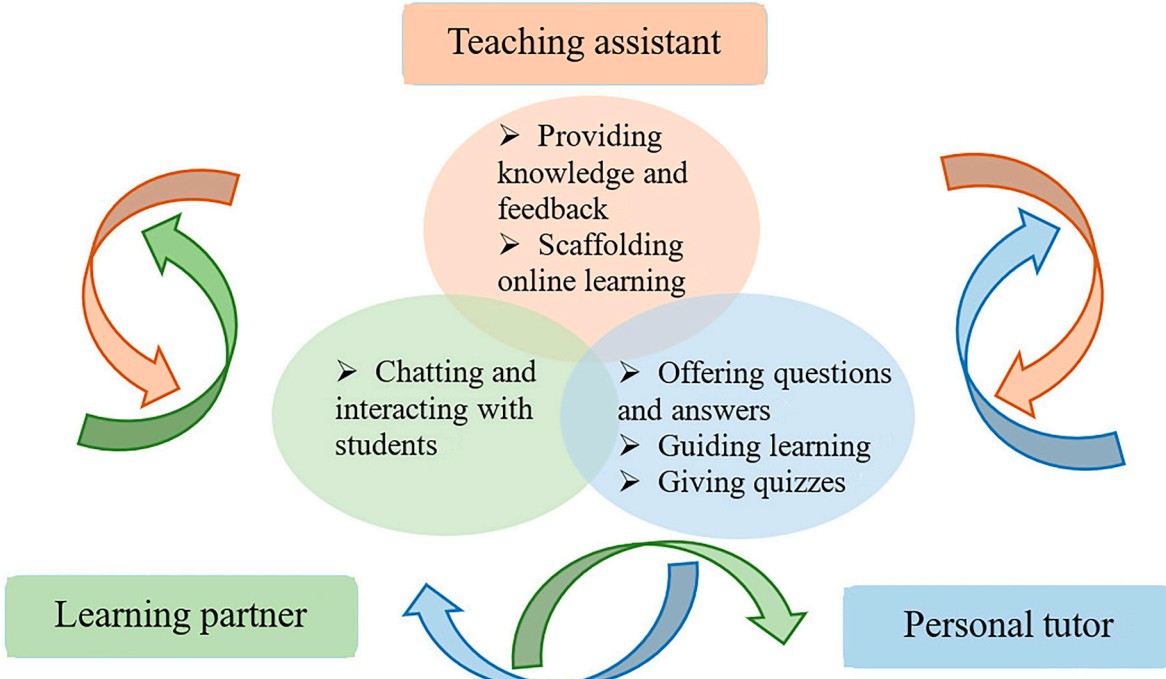

**Figure 1.** Three roles of chatbots in education.

### 2.2. Explicit Reasoning

Explicit reasoning is derived from the academically productive talk framework which emphasizes social interaction and reasoned participation [24]. The explicitness of reasoning is an important feature of students' effective communication. Peer dialogues should involve learning community, accurate knowledge, and rigorous thinking [25]. Specifically, students are expected to learn from each other, explicate their reasoning, and construct logical arguments in open-ended discussions and collaborative activities. It was found that chatbots could trigger and scaffold students' discussions, stimulating explicit reasoning processes [26]. Moreover, explicit reasoning could increase collaboration practices, improve learning outcomes, and promote conceptual knowledge acquisition [27]. The authors thus raised the following null hypothesis.

**H2.** *The use of chatbot technology could not significantly improve explicit reasoning at the 0.05 level.*

### 2.3. Learning Achievement and Knowledge Retention

Learning achievement is the measurement of students' academic success in a given period of time [28]. Most of the existing research on chatbot-based learning investigated learning achievement, including gains in second language speaking proficiency [29], aircraft engine maintenance test scores [30], vaccine knowledge scores [31], numerical system conversion test results [18], and transfer ability [32]. Through the pretest–posttest design, Ghanaian undergraduate students in the chatbot group better performed academically than those interacting with instructors [33]. However, chatbots may not significantly improve secondary school students' academic performance [3].

Knowledge retention, also known as learning retention, is defined as the ability to capture information and transfer it from short-term to long-term memory [34]. Many researchers mainly examined this ability via retention tests, e.g., delayed posttests for vocabulary knowledge [35] and multiple-choice question tests for programming knowledge [32]. The dynamic assessment of chatbots was effective in enhancing vocabulary retention and providing detailed information about personal learning [2]. The chatbot also facilitated the retention of Python programming knowledge [32]. Therefore, the authors proposed the following null hypotheses.

**H3.** *The use of chatbot technology could not significantly improve learning achievement at the 0.05 level.*

**H4.** *The use of chatbot technology could not significantly improve knowledge retention at the 0.05 level.*

*2.4. Learners' Engagement, Motivation, and Interest*

Many previous studies have examined the effect of chatbots on students' engagement. Engagement is the extent to which students actively involve or participate in learning activities [36]. A narrative-based learning system equipped with chatbot feedback significantly improved users' engagement [37]. Likewise, a mobile chatbot-based learning approach enabled nursing students to believe that chatbots could promote their learning engagement [31]. However, the interaction with chatbots failed to make a statistically significant difference in engagement in the extensive reading activity [38]. Considering inconsistent findings, the following null hypothesis was determined.

**H5.** *The use of chatbot technology could not significantly enhance learning engagement at the 0.05 level.*

Researchers also investigated the effect of chatbots on learning motivation. Motivation, unlike engagement, refers to the possibility of engaging in learning tasks and maintaining learning behavior [39]. Essentially, engagement highlighted action, while motivation emphasized intent [40]. Motivation can be categorized into intrinsic and extrinsic motivations [41]. Intrinsic motivation focuses on learners' inner satisfaction, whereas extrinsic motivation is defined as the behavior for external and separable results [42]. The voice-based chatbots positively influenced middle school students' motivation [43]. Furthermore, chatbot-assisted instructional videos and micro-learning systems could also effectively promote intrinsic motivation [18,44]. Thus, the authors presented the following null hypothesis.

**H6.** *The use of chatbot technology could not significantly enhance learning motivation at the 0.05 level.*

Previous studies also focused on the effect of chatbot technology on learning interest. Interest indicates the individual readiness or predisposition to engage in a given learning task with effort [45]. Chatbots improved Korean students' interest in foreign language learning [43]. Similarly, nursing college students using chatbots experienced a higher level of learning interest than their peers in the control group [46]. Chatbots also increased college students' interest in English vocabulary learning [47]. The authors thus raised the following null hypothesis.

**H7.** *The use of chatbot technology could not significantly enhance learning interest at the 0.05 level.*

*2.5. Intervention Duration*

Previous studies experimented with different durations and obtained different results. The study [18] whose intervention lasted for only 40 min found no significant differences in learning performance between the chatbot and control groups. Fifteen instructional sessions carried out over three weeks facilitated students' comprehension of English adjectival constructions but failed to help learners generate prepositional constructions [48]. However, speaking test scores in the experimental group were significantly higher than those of the control group after the four-month experiment [49]. In the field of educational technology, some meta-analyses have investigated the influence of implementation duration. For example, Chen et al. [50] examined the effects of mobile devices on language learning across different intervention durations. The authors therefore developed the following null hypothesis.

**H8.** *Intervention duration could not influence the effect of chatbot-assisted learning.*

### 2.6. Chatbot Roles

Researchers assigned different roles to chatbots in their experiments. University students created conversations with the chatbot Elbot ranging from school life to movies. After eight weeks, the experimental group better acquired vocabulary knowledge than the control group [47]. The AsasaraBot, acting as a tutor, provided questions, encouragement, and interactions, aiming to support students' language learning [15]. However, the experimental group performed worse than students equipped with other technological tools such as Google Forms. Using chatbots as teaching assistants, learners significantly outperformed those in traditional classroom settings in terms of projected-based learning performance [51]. The meta-analysis of robot-assisted language learning [52] examined the influence of robot roles on the effectiveness of social robots. Therefore, the authors formulated the following null hypothesis.

**H9.** *Chatbot roles could not influence the effect of chatbot-assisted learning.*

### 2.7. Learning Content

Chatbot technology has been applied to many disciplines; thus, participants' learning content has varied among studies. In nursing education, students used chatbot systems to learn courses about the physical examination, effectively enhancing students' academic performance [53]. The LINE Bot used in military science could provide procedures for engine fan module decomposition, which improved trainee performance and reduced training costs [30]. In the field of second language learning, students interacted with chatbots to practice their speaking skills [29]. However, the chatbot was ineffective when students developed computing knowledge, e.g., conversion of numerical systems [18]. The authors thus proposed the following null hypothesis.

**H10.** *Learning content could not influence the effect of chatbot-assisted learning.*

## 3. Materials and Methods

The researchers conducted a meta-analysis by collecting studies, coding included studies, and calculating the effect sizes. They strictly followed the Preferred Reporting Items for Systematic reviews and Meta-Analyses (PRISMA) guidelines [54]. It was not necessary to pre-register this systematic review in a designated public repository such as Prospero since this study did not involve the health of animals and human beings.

### 3.1. Literature Search

At the beginning, the authors determined search keywords by clustering the literature. They obtained 741 results in the Web of Science by keying in the research themes "chatbot" AND "learning". The results were output in the form of plain texts and imported into VOSviewer. The type of analysis was co-occurrence, and the unit of analysis was all keywords. The minimum number of occurrences of a keyword was set at three. Of the 2295 keywords, 217 met the threshold and were categorized into 11 clusters (Figure 2). The researchers obtained the top 10 frequently-occurring keywords: chatbot (N = 340), artificial intelligence (N = 96), machine learning (N = 90), chatbots (N = 89), deep learning (N = 81), natural language processing (N = 73), conversational agents (N = 36), conversational agent (N = 28), education (N = 24), and technology (N = 23).

The authors obtained 2322 studies from online databases on 3 September 2022. The major databases included Web of Science, Wiley Online Library, Springer Link, Taylor & Francis Online, Elsevier ScienceDirect, and Google Scholar (Figure 3). Considering the above findings of bibliometric analysis and the aim of this study, the researchers retrieved 228 results by entering the topic terms: (chatbot OR "conversational agent") AND (education OR learn* OR teach*) AND ("control group" OR experim* OR experient*) from Web of Science Core Collection. They also obtained 56 results from Wiley and 54 results from Taylor & Francis by keying in "chatbot OR conversational agent" AND "education

OR learning OR teaching OR control OR experiment" in the Abstract. Researchers obtained 20 records from Springer and 1860 results from Google Scholar by entering in "chatbot" in *where the title contains* AND "control group" in *with the exact phrase* AND "education OR learn OR teach" in *with at least one of the words*. Researchers also retrieved 104 studies by keying in "chatbot OR conversational agent" in the Title AND "education OR learning OR teaching OR control OR experiment" in Title, abstract, keywords from Elsevier.

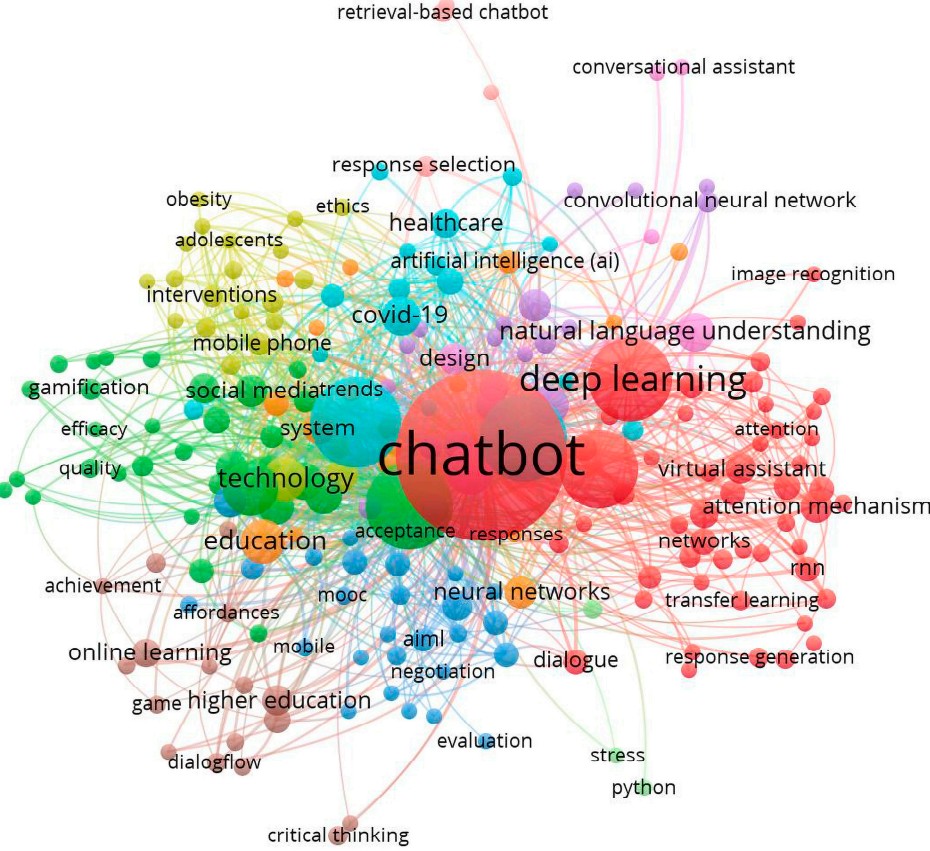

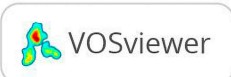

**Figure 2.** Clustering keywords related to chatbot-assisted learning.

### 3.2. Inclusion and Exclusion Criteria

The researchers followed the inclusion and exclusion criteria to select the literature. The identified studies should (1) determine the effect of chatbot technology on educational outcomes; (2) include the experimental group that adopted chatbot-based learning and the control group that used traditional learning approaches; (3) report sufficient statistics, i.e., sample sizes, means, and standard deviations, to calculate effect sizes; (4) ensure homogeneity between the control group and the experimental group, i.e., students' prior learning outcomes should be equivalent; and (5) be written in English and be published from 2010 to 2022. The studies were excluded if they (1) were irrelevant to educational use of chatbots; (2) lacked control groups; (3) did not provide adequate information for effect-size calculations; and (4) were not written in an acceptable English language.

Based on the inclusion and exclusion criteria, the researchers included 25 studies in this meta-analysis after the first round of screening. To avoid the case that some studies may be excluded by mistake, the researchers conducted another two rounds of screening. They found that four studies, which met all the inclusion criteria, were accidentally excluded in the first round of screening. Therefore, given three studies from previous reviews, there were altogether 32 studies included in this meta-analysis (Figure 3).

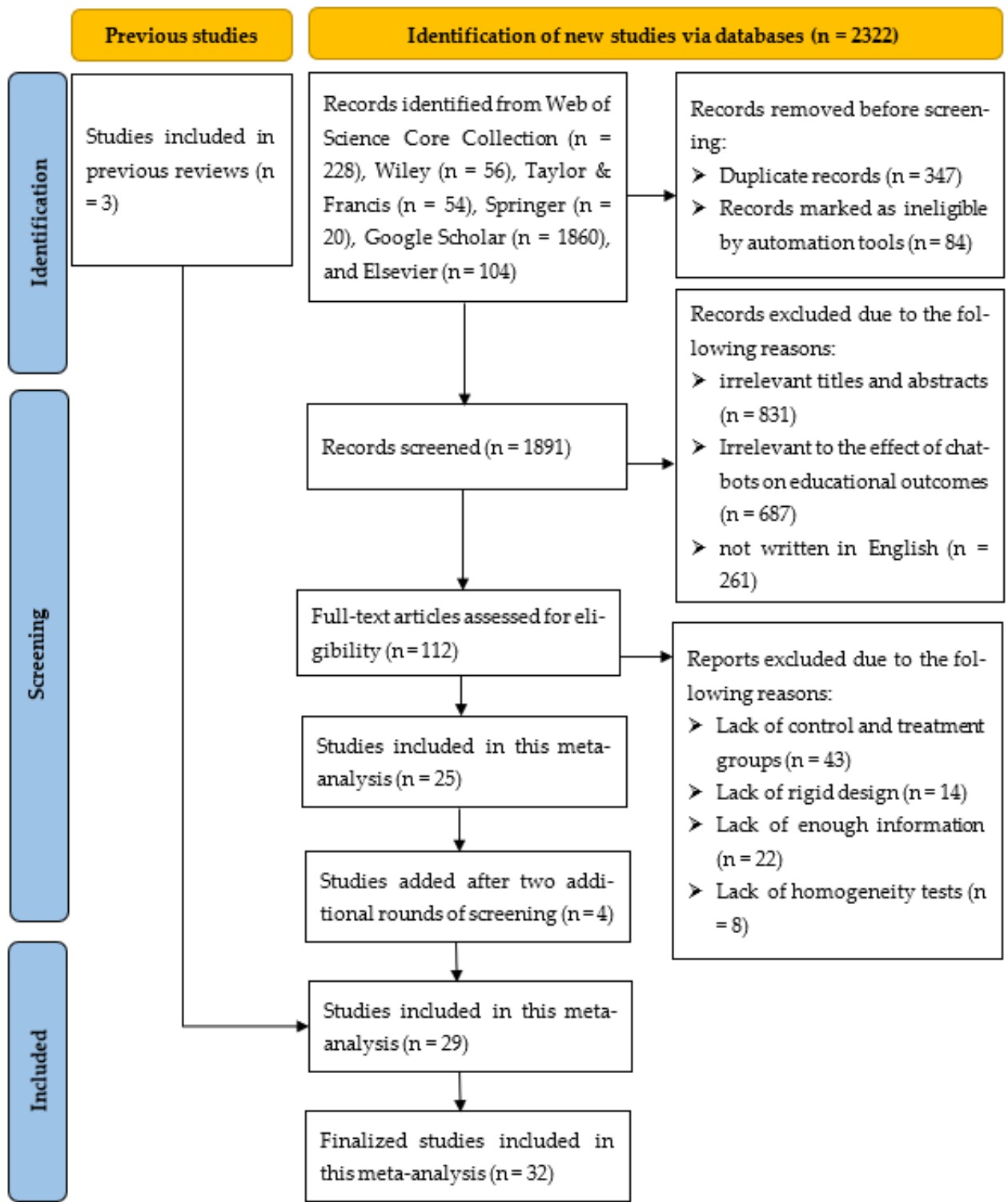

**Figure 3.** The flowchart of study inclusion.

*3.3. Coding Procedures*

The researchers developed a coding scheme consisting of comparable features of chatbot-based learning. First, basic information included the author's last name, publication year, and sample sizes. Second, regarding the instruction duration, the researchers at first coded it as a continuous variable. However, inspired by Chen et al. [50], the researchers decided to recode this variable as a categorical one for further analysis. Considering the data characteristics, the researchers divided the duration variable into five categories:

less than 1 week, less than 5 weeks, less than 10 weeks, more than 10 weeks, and not specified. Third, the chatbot role variable was classified into teaching assistants, tutors, and learning partners. Fourth, the researchers categorized the educational outcomes into critical thinking, explicit reasoning, learning achievement, knowledge retention, learning engagement, learning motivation, and learning interest. The defining terms of the above categories were, respectively, critical thinking scores [53], the frequency of explicit position and explicit argument [24], test scores [29], retention test scores [2], engagement scale scores [38], motivation questionnaire scores [18], and interest scale scores [47]. Fifth, in terms of learning content, the researchers divided it into five categories, i.e., computer science, instructional technology, language, medicine, and others. Table 2 shows the detailed information of included studies.

**Table 2.** The coding results of included studies.

| N | Author (Year) | Sample Size (Experimental Group/Control Group) | Instruction Duration | Chatbot Role | Learning Content | Educational Outcomes |
|---|---|---|---|---|---|---|
| 1 | Tegos and Demetriadis (2017) [26] | 72 (38/34) | 40 min | Tutor | Computer science | Explicit reasoning and learning achievement |
| 2 | Winkler et al. (2020) [32] | 72 (37/35) | 30 min | Tutor | Computer science | Learning achievement and retention |
| 3 | Song and Kim (2021) [16] | 56 (27/29) | 15 weeks | Partner | Instructional technology | Learning achievement |
| 4 | Chang et al. (2022) [53] | 32 (16/16) | 100 min | Tutor | Medicine | Critical thinking and learning achievement |
| 5 | Tegos et al. (2015) [55] | 43 (21/22) | 70 min | Tutor | Computer science | Explicit reasoning and learning achievement |
| 6 | Tegos et al. (2016) [24] | 64 (32/32) | 40 min | Tutor | Computer science | Explicit reasoning and learning achievement |
| 7 | Fidan and Gencel (2022) [44] | 94 (54/40) | 4 weeks | Tutor | Instructional technology | Learning motivation and achievement |
| 8 | H. L. Chen et al. (2020) [17] | 58 (19/29) | 4 weeks | Tutor | Language | Learning achievement and retention |
| 9 | Yuan et al. (2021) [30] | 40 (20/20) | 2–3 weeks | Tutor | Others | Learning achievement |
| 10 | Mageira et al. (2022) [15] | 35 (18/17) | 2 days | Tutor | Language | Learning achievement |
| 11 | Vazquez-Cano et al. (2021) [56] | 103 (52/51) | 2 weeks | Teaching assistant | Language | Learning achievement |
| 12 | Chang, Hwang, et al. (2022) [31] | 36 (18/18) | 100 min | Tutor | Medicine | Learning achievement |
| 13 | Yin et al. (2021) [18] | 99 (51/48) | 40 min | Tutor | Computer science | Learning motivation and achievement |
| 14 | Lee et al. (2022) [57] | 38 (18/20) | 200 min | Tutor | Medicine | Learning motivation and achievement |
| 15 | Ruan et al. (2020) [37] | 36 (18/18) | 25.75 min | Tutor | Others | Learning engagement and achievement |
| 16 | Kim (2018) [47] | 47 (24/23) | 8 weeks | Partner | Language | Learning interest, motivation, and achievement |

**Table 2.** *Cont.*

| N | Author (Year) | Sample Size (Experimental Group/Control Group) | Instruction Duration | Chatbot Role | Learning Content | Educational Outcomes |
|---|---|---|---|---|---|---|
| 17 | Han (2020) [43] | 44 (22/22) | 10 weeks | Partner | Language | Learning interest, motivation, and achievement |
| 18 | Kim (2018) [58] | 46 (24/22) | 16 weeks | Partner | Language | Learning achievement |
| 19 | Kumar (2021) [51] | 60 (30/30) | 10 weeks | Teaching assistant | Instructional technology | Learning achievement |
| 20 | Kim et al. (2021) [59] | 75 (37/38) | 1 semester | Partner | Language | Learning achievement |
| 21 | Jeon (2021) [2] | 35 (18/17) | 25 min | Tutor | Language | Learning achievement and retention |
| 22 | Farah et al. (2022) [60] | 20 (11/9) | 30 min | Tutor | Computer science | Learning engagement and achievement |
| 23 | Goda et al. (2014) [61] | 63 (31/32) | 30 min | Partner | Language | Critical thinking and learning achievement |
| | | 67 (32/35) | 30 min | Partner | Language | Critical thinking |
| 24 | Wambsganss et al. (2021) [62] | 55 (31/24) | 15 min | Teaching assistant | Language | Learning achievement |
| 25 | Kim (2022) [48] | 64 (32/32) | 3 weeks | Tutor | Language | Learning achievement and retention |
| 26 | Dizon (2020) [29] | 28 (13/15) | 10 weeks | Tutor | Language | Learning achievement |
| 27 | Abbasi et al. (2019) [63] | 110 (55/55) | Not available | Teaching assistant | Computer science | Learning achievement |
| 28 | Abbasi and Kazi (2014) [64] | 72 (36/36) | Not available | Teaching assistant | Computer science | Learning achievement |
| 29 | Lin and Chang (2020) [65] | 357 (167/190) | 2 weeks | Teaching assistant | Language | Learning achievement |
| 30 | Liu et al. (2022) [38] | 62 (41/21) | 6 weeks | Partner | Language | Learning engagement and interest |
| 31 | Hsu et al. (2021) [49] | 48 (24/24) | 4 months | Tutor | Language | Learning achievement |
| 32 | Na-Young (2019) [66] | 70 (36/34) | 10 sessions | Partner | Language | Learning achievement |

### 3.4. Data Analysis

This study used Stata MP/14.0 to carry out the meta-analysis. Cohen's *d*, responsible for the measurement of effect sizes, is calculated by dividing the mean difference between the experimental and control groups by the pooled standard deviation [67]. The calculation formula is as follows. The researchers calculated 76 effect sizes with a total sample size of 2201 in 32 identified studies.

$$\text{Cohen's } d = \frac{M_E - M_C}{\sqrt{\frac{(N_E - 1)\,S_E^2 + (N_C - 1)\,S_C^2}{(N_E - 1) + (N_C - 1)}}} \tag{1}$$

The researchers examined publication bias using both visual and mathematical tests, including the funnel plot, Begg's test, and Egger's test. Through the shape of the funnel plot, researchers could preliminarily assess publication bias. The dots will be symmetrically

distributed along the no-effect line if there is no publication bias, while the plot will be asymmetric if there is a presence of publication bias. Statistically, both Begg's and Egger's tests reported *p*-value, which determines the presence or absence of publication bias.

The researchers also measured heterogeneity using $I^2$ test and conducted sensitivity analysis. The heterogeneity will be considered low if $I^2$ is less than 25%, moderate if $I^2$ falls between 25% and 75%, and substantial if $I^2$ is greater than 75% [68]. Generally, heterogeneity is significant when $I^2$ is larger than 50% and a random effects model should be adopted accordingly. Otherwise, heterogeneity is insignificant, and the fixed effects model should be used.

## 4. Results

### 4.1. Analyses of Publication Bias

Both Egger's and Begg's tests (Table 3) checked whether the identified studies were influenced by publication bias. The results of both tests indicated the absence of publication bias in critical thinking ($t = 14.17$, $p = 0.111$; $z = 1.47$, $p = 0.142$), explicit reasoning ($t = 1.90$, $p = 0.424$; $z = 1.32$, $p = 0.188$), learning achievement ($t = 4.36$, $p = 0.083$; $z = 0.53$, $p = 0.598$), knowledge retention ($t = 10.18$, $p = 0.061$; $z = 1.47$, $p = 0.142$), learning engagement ($t = -0.14$, $p = 0.946$; $z = 1.35$, $p = 0.176$), and learning motivation ($t = 8.17$, $p = 0.216$; $z = 1.23$, $p = 0.217$). The funnel plot also confirmed test results (Figure 4). Using learning motivation as an example, the funnel graph was obviously symmetrical, revealing no publication bias. However, Begg's test showed a presence of publication bias in learning interest ($z = 2.04$, $p = 0.042$), which was different from the result of Egger's test ($t = 15.30$, $p = 0.056$).

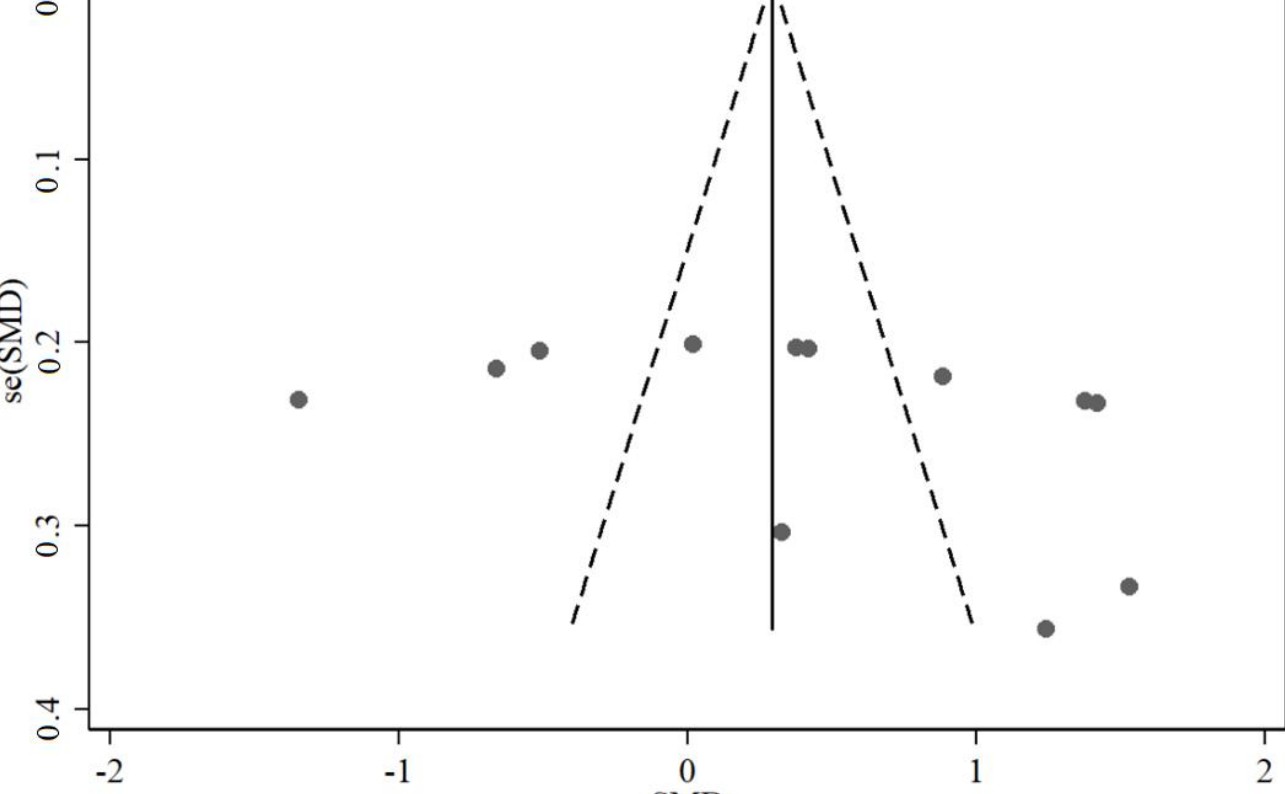

**Figure 4.** A funnel plot of publication bias of learning motivation.

**Table 3.** Test results of publication bias.

| N | Outcome | n | Egger's Test | | Begg's Test | | | Continuity Corrected | |
|---|---|---|---|---|---|---|---|---|---|
| | | | Bias | *p* | Score | sd | *z* | *p* | *z* | *p* |
| 1 | Critical thinking | 5 | 14.17 | 0.111 | 6 | 4.08 | 1.47 | 0.142 | 1.22 | 0.221 |
| 2 | Explicit reasoning | 6 | 1.90 | 0.424 | 7 | 5.32 | 1.32 | 0.188 | 1.13 | 0.260 |
| 3 | Learning achievement | 33 | 4.36 | 0.083 | 34 | 64.54 | 0.53 | 0.598 | 0.51 | 0.609 |
| 4 | Knowledge retention | 5 | 10.18 | 0.061 | 6 | 4.08 | 1.47 | 0.142 | 1.22 | 0.221 |
| 5 | Learning engagement | 7 | −0.14 | 0.946 | 9 | 6.66 | 1.35 | 0.176 | 1.20 | 0.230 |
| 6 | Learning motivation | 12 | 8.17 | 0.216 | 18 | 14.58 | 1.23 | 0.217 | 1.17 | 0.244 |
| 7 | Learning interest | 4 | 15.30 | 0.056 | 6 | 2.94 | 2.04 | 0.042 | 1.70 | 0.089 |

*4.2. Results of Sensitivity Analysis*

The researchers implemented a sensitivity analysis to examine the stability of meta-analytical results. As shown in Figure 5, all estimates range from the lower confidence interval limit (95% CI = 0.55) to the upper confidence interval limit (95% CI = 0.68). It indicates that none of the included studies could influence the pooled effect size when a specific study is omitted, confirming that the meta-analytical results are robust.

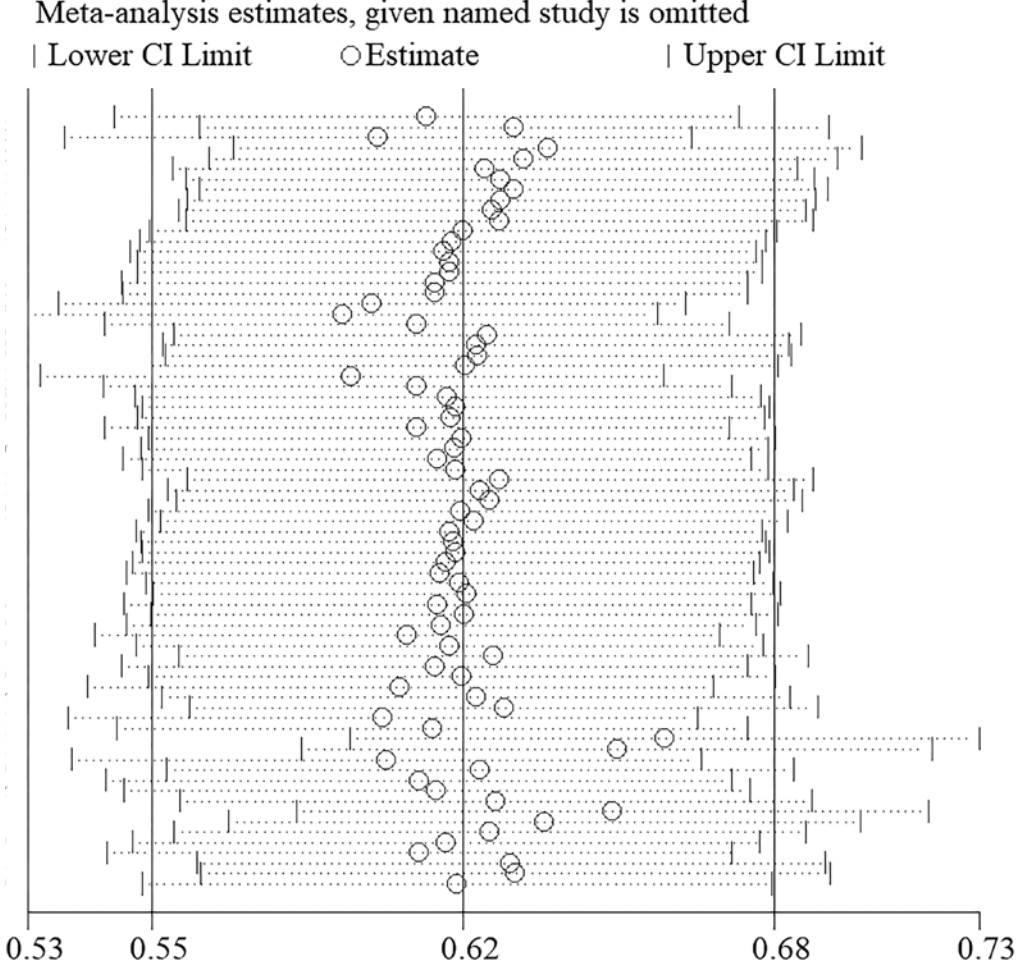

**Figure 5.** A plot of sensitivity analysis.

*4.3. Results of Moderator Analyses*

Table 4 presents the overall effect sizes of each outcome domain. Since $I^2$ statistics revealed that the effect size in critical thinking (Q = 55.89, $I^2$ = 92.8%, *p* < 0.001) was of considerable heterogeneity, the researchers adopted the random effects model. The effect

size did not show a significant difference in critical thinking ($d$ = 0.690, 95% CI [−0.235, 1.615], $p$ = 0.144) between the chatbot-assisted and conventional learning methods. Thus, the authors accepted hypothesis 1.

**Table 4.** The effect sizes of seven educational outcomes.

| N | Outcome | Effect Size | | | | Heterogeneity Test | | | |
|---|---------|-------------|---|---|---|--------------------|---|---|---|
| | | $d$ | 95% CI | z | $p$ | Q-Value | df | $p$ | $I^2$ |
| 1 | Critical thinking | 0.690 | −0.235, 1.615 | 1.46 | 0.144 | 55.89 | 4 | 0.000 | 92.8% |
| 2 | Explicit reasoning | 1.190 | 0.868, 1.512 | 7.25 | 0.000 | 1.88 | 5 | 0.865 | 0.0% |
| 3 | Learning achievement | 1.033 | 0.743, 1.322 | 6.99 | 0.000 | 221.14 | 32 | 0.000 | 85.5% |
| 4 | Knowledge retention | 0.691 | 0.101, 1.281 | 2.29 | 0.022 | 22.96 | 4 | 0.000 | 82.6% |
| 5 | Learning engagement | 0.147 | −0.068, 0.363 | 1.34 | 0.180 | 3.22 | 6 | 0.780 | 0.0% |
| 6 | Learning motivation | 0.409 | −0.099, 0.916 | 1.58 | 0.114 | 161.61 | 11 | 0.000 | 93.2% |
| 7 | Learning interest | 0.842 | 0.034, 1.650 | 2.04 | 0.041 | 23.39 | 3 | 0.000 | 87.2% |
| 8 | Overall | 0.789 | 0.593, 0.985 | 7.90 | 0.000 | 582.09 | 71 | 0.000 | 87.8% |

However, the effect size in explicit reasoning (Q = 1.88, $I^2$ = 0.0%, $p$ = 0.865) was not significantly heterogeneous. The fixed effects model was thus used. Table 4 shows a significant difference in explicit reasoning ($d$ = 1.190, 95% CI [0.868, 1.512], $p$ < 0.001) between the experimental and control groups. The authors thus rejected hypothesis 2.

Effect sizes in both learning achievement (Q = 221.14, $I^2$ = 85.5%, $p$ < 0.001) and knowledge retention (Q = 22.96, $I^2$ = 82.6%, $p$ < 0.001) were considered heterogeneous. The authors accordingly employed the random effects model to pool the data. It was revealed that learning achievement ($d$ = 1.033, 95% CI [0.743, 1.322], $p$ < 0.001) and retention ($d$ = 0.691, 95% CI [0.101, 1.281], $p$ = 0.022) in the experimental group significantly improved compared with the control group. Therefore, the authors rejected hypotheses 3 and 4.

Because of $I^2$ = 0.0% in learning engagement (Q = 3.22), the effect sizes were of insignificant heterogeneity at the 0.05 level ($p$ = 0.780). The authors thus chose the fixed effects model to pool the effect sizes. No significant differences in learning engagement ($d$ = 0.147, 95% CI [−0.068, 0.363], $p$ = 0.114) were found between chatbot-assisted and traditional learning methods. The authors, therefore, accepted hypothesis 5.

The effect sizes in learning motivation (Q = 161.61, $I^2$ = 93.2%, $p$ < 0.001) and interest (Q = 23.39, $I^2$ = 87.2%, $p$ < 0.001) were heterogeneous. Thus, the authors adopted the random effects model when conducting the meta-analysis regarding both educational outcomes. Meta-analytical results (Table 4) showed that compared with the control group, the experimental group maintained significantly higher levels of learning interest ($d$ = 0.842, 95% CI [0.034, 1.650], $p$ = 0.041). However, learning motivation in both groups was not significantly different ($d$ = 0.409, 95% CI [−0.099, 0.916], $p$ = 0.114). The authors thus accepted hypothesis 6 and rejected hypothesis 7.

The overall effect size of using chatbots in education was 0.789 ($p$ < 0.001), with a 95% confidence interval between 0.593 and 0.985 (Table 4). According to Cohen's [69] effect-size criteria, which identified 0.2, 0.5, and 0.8 as small, medium, and large effect sizes respectively. The overall effect size in this study was thus considered as a medium-to-large level, indicating that the use of chatbot technology exerted a positive effect on learning outcomes. Additionally, the heterogeneity test showed that the overall effect size (Q = 582.09, $I^2$ = 87.8%, $p$ < 0.001) was of considerable heterogeneity, which deserved further analysis to explore potentially critical moderator variables.

The researchers implemented a meta-regression analysis for three moderator variables (Table 5). Regarding intervention duration, there were medium-sized effects on chatbot-assisted learning for less than one week ($d$ = 0.775, $p$ < 0.001), less than 10 weeks ($d$ = 0. 561, $p$ < 0.001), and more than 10 weeks ($d$ = 0.601, $p$ < 0.01) and large-sized effects for less than 5 weeks ($d$ = 1.060, $p$ < 0.05) and not specified ($d$ = 1.844, $p$ < 0.001). However, there were no significant differences between effect sizes of the different intervention durations ($p$ > 0.05). Thus, hypothesis 8 was accepted.

**Table 5.** The effect sizes of moderator variables.

| N | Moderator Variable | Effect Size | | | | | Heterogeneity Test | | | |
|---|---|---|---|---|---|---|---|---|---|---|
| | | *n* | *d* | 95% CI | *z* | *p* | Coefficient | 95% CI | *t* | *p* |
| 1 | Intervention duration | | | | | | 0.026 | −0.194, 0.247 | 0.24 | 0.814 |
| | Less than 1 week | 39 | 0.775 | 0.561, 0.990 | 7.09 | 0.000 | | | | |
| | Less than 5 weeks | 11 | 1.060 | 0.218, 1.902 | 2.47 | 0.014 | | | | |
| | Less than 10 weeks | 18 | 0.561 | 0.318, 0.805 | 4.52 | 0.000 | | | | |
| | More than 10 weeks | 2 | 0.601 | 0.203, 0.998 | 2.96 | 0.003 | | | | |
| | Not specified | 2 | 1.844 | 1.496, 2.193 | 10.37 | 0.000 | | | | |
| 2 | Chatbot roles | | | | | | −0.012 | −0.271, 0.248 | −0.09 | 0.930 |
| | Teaching assistant | 4 | 1.631 | 1.012, 2.250 | 5.17 | 0.000 | | | | |
| | Tutor | 46 | 0.748 | 0.491, 1.005 | 5.71 | 0.000 | | | | |
| | Partner | 22 | 0.712 | 0.438, 0.986 | 5.09 | 0.000 | | | | |
| 3 | Learning content | | | | | | 0.061 | −0.147, 0.270 | 0.59 | 0.559 |
| | Computer science | 24 | 0.695 | 0.267, 1.123 | 3.18 | 0.001 | | | | |
| | Instructional technology | 8 | 0.928 | 0.642, 1.215 | 6.35 | 0.000 | | | | |
| | Language | 33 | 0.749 | 0.546, 0.951 | 7.25 | 0.000 | | | | |
| | Medicine | 4 | 1.588 | 0.363, 2.812 | 2.54 | 0.011 | | | | |
| | Others | 3 | 0.519 | −0.422, 1.461 | 1.08 | 0.279 | | | | |

The results for the remaining two variables presented similar patterns. Regarding chatbot roles, a large effect size was reported for using chatbots as teaching assistants ($d = 1.631$, $p < 0.001$), while the role of tutors ($d = 0.748$, $p < 0.001$) and partners ($d = 0.712$, $p < 0.001$) yielded medium effect sizes. Nevertheless, no significant differences existed between effect sizes of three roles of chatbots ($p > 0.05$). The authors thus accepted hypothesis 9.

In terms of learning content, computer science ($d = 0.695$, $p < 0.01$) and language ($d = 0.749$, $p < 0.01$) showed medium effect sizes, and large effect size estimates came from instructional technology ($d = 0.928$, $p < 0.001$) and medicine ($d = 1.588$, $p < 0.05$). The categories of math and military science were merged into one category (i.e., others) due to the limited studies, with no significant effect sizes ($d = 0.519$, $p > 0.05$). The meta-regression results revealed no statistically significant differences between the abovementioned values. Therefore, the authors accepted hypothesis 10. The results of the hypothesis testing are summarized in Table 6.

**Table 6.** The results of hypothesis testing.

| N | Null Hypotheses | Results |
|---|---|---|
| 1 | The use of chatbot technology could not significantly improve critical thinking at the 0.05 level. | Accepted |
| 2 | The use of chatbot technology could not significantly improve explicit reasoning at the 0.05 level. | Rejected |
| 3 | The use of chatbot technology could not significantly improve learning achievement at the 0.05 level. | Rejected |
| 4 | The use of chatbot technology could not significantly improve knowledge retention at the 0.05 level. | Rejected |
| 5 | The use of chatbot technology could not significantly enhance learning engagement at the 0.05 level. | Accepted |
| 6 | The use of chatbot technology could not significantly enhance learning motivation at the 0.05 level. | Accepted |
| 7 | The use of chatbot technology could not significantly enhance learning interest at the 0.05 level. | Rejected |
| 8 | Intervention duration could not influence the effect of chatbot-assisted learning. | Accepted |
| 9 | Chatbot roles could not influence the effect of chatbot-assisted learning. | Accepted |
| 10 | Learning content could not influence the effect of chatbot-assisted learning. | Accepted |

## 5. Discussion

This study investigated the effect of chatbot-assisted learning on various components and how different moderator variables influenced the effectiveness. There were no significant changes in critical thinking through the use of chatbots, which was inconsistent with the findings of Li et al. [13]. Although limited studies focused on critical thinking, there were still contradictory findings, probably because of the elusive property and different measurement instruments. Critical thinking is difficult to measure. Some studies

(e.g., [53]) used a newly developed scale consisting of five items. However, other studies (e.g., [61]) adopted the four-factor inventory developed through exploratory factor analysis. Chatbots may only exert influence on some dimensions of critical thinking, such as inquiring mindset.

The use of chatbot technology significantly enhanced explicit reasoning, which was also underexplored. The existing studies mainly developed students' explicit reasoning in collaborative activities since one display of explicit reasoning could associate with another one, which was the core of transactivity theory [70]. Chatbot interventions could effectively stimulate group discussions and help students utter their thoughts [55]. Chatbots also asked for clear and convincing statements, motivating students to find strong evidence. Thus, the conversational behaviors in the chatbot group were considered more transactive, i.e., with more explicit arguments, than those in the control group.

Chatbot technology also significantly improved learning achievement and retention. This finding was corroborated by the previous studies that confirmed the positive influence of chatbots on linguistic competence [10] and programming course achievement [33]. Chatbots can easily notice learners' knowledge gaps and accordingly make responses in order to create meaningful interactions. Review activities before presenting new information activate students' prior knowledge, facilitating the integration of the new and old knowledge [57]. Regarding knowledge retention, chatbots could randomly generate multiple-choice questions for declarative knowledge testing and open questions for procedural knowledge testing. Students in this way could timely recall their newly acquired information.

However, significant differences in learning engagement and motivation were not found between the chatbot-based condition and the control condition. One possible reason was that some students preferred to finish learning tasks in their own ways and paid little attention to chatbots [38], leading to a decrease in learning engagement. Another reason may be that factors such as peer feedback could influence motivation and that influencing factors varied with learning environments. Specifically, pressure was a significant predictor of motivation in the chatbot-assisted learning context, whereas perceived competency was an influencing factor in the traditional context [18].

With chatbot technology, students experienced more learning interest than those without it, supporting previous studies [10,43,46]. The enhancement of learning interest can be attributed to the flexibility of learning and affective feedback. Chatbot systems allowed users to learn based on individual needs and pace, which avoided frustration and learned helplessness for slow learners. Chatbots were designed to give encouraging messages if students failed to correctly answer questions [30]. They also gave human-like utterances such as *uh-huh* and *yeah*.

Three types of chatbot roles revealed no significant differences in learning outcomes. This finding echoed Huang et al.'s [1] suggestion for future research on determining how chatbots can be utilized to best achieve learning outcomes. Students can benefit from chatbot technology regardless of its roles. Chatbots were qualified as teaching assistants and learning partners since human teachers still took the leading role. Interestingly, chatbots could also be employed as tireless personal tutors. The intelligent systems guided and monitored students' personalized learning, while teachers had the opportunity and time to discover learners' potential problems [53], thus jointly promoting learning outcomes.

Intervention duration failed to influence the effectiveness of chatbot-based learning. The result did not indicate the novelty effect that learning outcomes may improve in the short term but ultimately decrease over time, which was inconsistent with the study [8]. It was possibly because a growing number of studies (e.g., [44,55]) have attempted to mitigate the novelty effect by introducing and familiarizing students with chatbot technology prior to intervention. Students in the information age can readily reach different technological innovations. Therefore, they were most likely to familiarize themselves with chatbots within several minutes.

Learning content was also not a significant variable. Moderator analysis suggested a more positive result for computer science, instructional technology, language, and medicine than the "others" category. Computer science and language were the most targeted fields in chatbot-based education, while engineering and mathematics received less attention [12]. Due to the small number of studies on other domains, educational fields such as military science and mathematics were subsumed into one category in this study. Therefore, it cannot be concluded that chatbots were more effective for certain learning content than for other categories.

## 6. Conclusions

### 6.1. Major Contributions

Methodologically, this study included major databases and conducted a meta-analysis under the PRISMA guidelines to examine the effectiveness of chatbot technology on educational outcomes. Theoretically, the results showed a medium-to-high overall effect size of chatbots on educational outcomes regardless of intervention duration, chatbot roles, and learning content. Chatbot technology exerted a significant and positive influence on explicit reasoning, learning achievement, knowledge retention, and learning interest. However, chatbots did not significantly improve critical thinking, learning engagement, and motivation. Practically, teachers and instructors could adopt appropriate teaching approaches to facilitate sustainable education.

### 6.2. Limitations

It should be noted that there are some limitations to the present study. First, the researchers only included studies written in English. Some publications, especially in Korean, could not be understood by the researchers. Second, this study only included three moderator variables that did not significantly influence the effectiveness of chatbots. Third, the results may still be influenced by unpublished studies with insignificant results despite the absence of publication bias.

### 6.3. Implications for Future Research

The findings may shed light on future research directions and propose suggestions for practitioners. First, the results revealed that chatbot-based learning was more effective than traditional learning in terms of explicit reasoning, learning achievement, knowledge retention, and learning interest. Therefore, future research could explore more components of educational outcomes, e.g., learning confidence, self-efficacy, social media use, and cognitive load. Experimental studies with large sample sizes are also expected. Teachers and instructors could integrate chatbot technology with different activities to meet learners' needs. Educational institutions could also provide training to improve teachers' and students' digital literacy and knowledge about artificial intelligence [71].

Second, future research could further explore users' attitudes towards chatbot technology and students' learning attitudes. Since the control group in the included studies did not obtain access to chatbots, it was difficult to compare users' attitudes between the control and intervention groups. Researchers could employ such models as the technology acceptance model and task-technology fit model to analyze the influencing factors of users' attitudes. On the other hand, learning attitudes could be compared. Only a few studies, however, focused on this aspect [43,57]. Future research could also expand chatbot research by using interdisciplinary research methods.

Third, intervention duration, chatbot roles, and learning content did not influence learning effectiveness. Thus, researchers could include more potential moderator variables for future meta-analyses, e.g., educational levels and interaction types. Future studies could also consider chatbot integration in other underexplored disciplines, e.g., arts, mathematics, and psychology [12]. Teachers could feel free to adopt chatbot-integrated teaching. They could introduce chatbot technology at any stage of the semester and assign any role to chatbots according to teaching needs. Developers and designers could introduce intriguing

elements by learning natural language processing and improve chatbots' performance based on machine learning algorithms.

**Supplementary Materials:** The following supporting information can be downloaded at: https://www.mdpi.com/article/10.3390/su15042940/s1, File S1: Supporting data; File S2: PRISMA checklist.

**Author Contributions:** Conceptualization, X.D. and Z.Y.; methodology, X.D.; software, X.D. and Z.Y.; validation, X.D. and Z.Y.; formal analysis, X.D. and Z.Y.; investigation, X.D.; resources, X.D.; data curation, X.D.; writing—original draft preparation, X.D.; writing—review and editing, Z.Y.; visualization, X.D.; supervision, Z.Y.; project administration, Z.Y.; funding acquisition, X.D. and Z.Y. All authors have read and agreed to the published version of the manuscript.

**Funding:** This research was funded by the 2019 MOOC of Beijing Language and Culture University (MOOC201902) (important) "Introduction to Linguistics"; "Introduction to Linguistics" of online and offline mixed courses in Beijing Language and Culture University in 2020; the Special fund of Beijing Co-construction Project Research and reform of the "Undergraduate Teaching Reform and Innovation Project" of Beijing higher education in 2020-innovative "multilingual +" excellent talent training system (202010032003); The research project of Graduate Students of Beijing Language and Culture University "Xi Jinping: The Governance of China" (SJTS202108); The Fundamental Research Funds for the Central Universities, and the Research Funds of Beijing Language and Culture University (22YCX017).

**Institutional Review Board Statement:** Not applicable.

**Informed Consent Statement:** Not applicable.

**Data Availability Statement:** The data presented in this study are available in File S1 in Supplementary Materials.

**Acknowledgments:** The authors would like to extend their gratitude to the people who help this study and the projects which financially support this study.

**Conflicts of Interest:** The authors declare no conflict of interest.

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
