# Peer review of "A Meta-Analysis and Systematic Review of the Effect of Chatbot Technology Use in Sustainable Education"

_sustainability, doi:10.3390/su15042940_

Round 1
Reviewer 1 Report
It is an interesting review study which is well-developed and well-organized. The following points should be considered and corrected by the athor/s.
1. in Figure 2, studies included in meta-analysis is equal to 25; after two additional rounds, another 4 articles added= 29, but the final studies included is equal to 32. what are 3 additional articles and when have they been added?
2. the author is recommended to add a paragraph discussing the problems in educational setting where the study has beeen conducted, China and explain how this study can benefit Chinese educational system. It would be beneficial if the author tries to elaborate what motivated him/her to conduct such a study.
3. Although several hypotheses were stated in the study but no EXPLICIT reference was made to the hypotheses in results and discussion section to say if they were rejected or accepted.
Reviewer 2 Report
Many thanks for submitting your manuscript to Sustainability
Here and now, the conduction of the current study is appropriate and timing. However, some issues can be addressed in future submissions/revisions.
The results are solid, which offers warrants for the quality of the study.
The most critical issue is the literature/hypotheses. The literature needs to be revised/expanded/updated (at least doubled) to form the many hypotheses tested by the current study. The results rejected almost 40% of the hypotheses, which might be caused by the false hypotheses proposed.
The significance of the study needs to be articulated explicitly, especially for potential international readers.
The discussion must be more comprehensive to engage the study in a meaningful conversation with past scholarly works to outshine the study's contributions to the knowledge and field. Thus, I suggest the authors add more literature in the literature review and discussion sections.
The contributions of the study (methodological, theoretical, and practical contributions) need to be articulated explicitly.
Reviewer 3 Report
The paper is well-written by showing a clear structure and a solid study background.
Only two minor editing revisions as follows:
1. Insert a space between Table 2 and the coming "3.4 Data analysis” sub-paragraph at line 214.
2. Insert a space between Table 4 and the text below at line 261.
Reviewer 4 Report
The document is a relevant state of the art. A bibliometric analysis with VosViewer and Open Knowledge Maps would be advisable to attach.
On the other hand, it is suggested that the authors in the references section use a range between 2023 and 2019. Exclude conferences and verify articles from ScienceDirect, IEEE Xplore, Wiley, Taylor & Francis, PLOS, Springer, and Hindawi over MDPI (Education Sciences).
Round 2
Reviewer 2 Report
The authors have addressed all the comments offered in the previous review. Thus, I suggest accepting the article.
Author Response
Summary: The authors have addressed all the comments offered in the previous review. Thus, I suggest accepting the article.
Response: Thank you very much for your patience and appreciation.
